# Advances in MXene-Based Electrochemical (Bio)Sensors for Neurotransmitter Detection

**DOI:** 10.3390/mi14051088

**Published:** 2023-05-21

**Authors:** Meiqing Yang, Lu Wang, Haozi Lu, Qizhi Dong

**Affiliations:** 1Zoology Key Laboratory of Hunan Higher Education, College of Life and Environmental Science, Hunan University of Arts and Science, Changde 415000, China; 2Institute of Chemical Biology and Nanomedicine (ICBN), State Key Laboratory of Chemo/Biosensing and Chemometrics, College of Chemistry and Chemical Engineering, Hunan University, Changsha 410082, China

**Keywords:** electrochemical neurotransmitter sensors, MXene-based electrode materials, biogenic amines, amino acids, soluble gases

## Abstract

Neurotransmitters are chemical messengers that play an important role in the nervous system’s control of the body’s physiological state and behaviour. Abnormal levels of neurotransmitters are closely associated with some mental disorders. Therefore, accurate analysis of neurotransmitters is of great clinical importance. Electrochemical sensors have shown bright application prospects in the detection of neurotransmitters. In recent years, MXene has been increasingly used to prepare electrode materials for fabricating electrochemical neurotransmitter sensors due to its excellent physicochemical properties. This paper systematically introduces the advances in MXene-based electrochemical (bio)sensors for the detection of neurotransmitters (including dopamine, serotonin, epinephrine, norepinephrine, tyrosine, NO, and H_2_S), with a focus on their strategies for improving the electrochemical properties of MXene-based electrode materials, and provides the current challenges and future prospects for MXene-based electrochemical neurotransmitter sensors.

## 1. Introduction

Our human physiological state and behaviour are controlled by the nervous system. Specific signals generated by the nervous system are transmitted via synapses from one neuron to another neuron, muscle cell, or glandular cell, and ultimately cause a response in the body. Neurotransmitters, as endogenous chemical messengers that transmit signals across synapses, are essential for information exchange between the nervous system and the body [1,2,3]. To date, more than one hundred neurotransmitters have been identified. Depending on their function, neurotransmitters can be excitatory or inhibitory [4]. Based on their chemical structure, neurotransmitters can be divided into the following categories: biogenic amines, including dopamine (DA), serotonin (5-hydroxytryptamine [5-HT]), epinephrine (EP), norepinephrine (NE); amino acids, such as glutamate (Glu), γ-aminobutyric acid (GABA), aspartate (Asp), and tyrosine (Tyr); acetylcholine (Ach); and soluble gases, such as nitric oxide (NO) and hydrogen sulfide (H_2_S) [3,5]. Proper concentrations of neurotransmitters in the nervous system are critical for the human body. They not only control our physiological states (e.g., heart rate, blood pressure and muscle tone) but also regulate our daily behaviors (e.g., memory, sleep, mood, appetite, attention). Abnormal levels of neurotransmitters have been reported to be closely associated with some mental disorders (Alzheimer’s disease, Parkinson’s disease, autism, epilepsy, schizophrenia, depression) and physical illnesses (glaucoma, thyroid hormone deficiency, congestive heart injury) [2,6,7]. Hence, the accurate quantification of neurotransmitter levels in the body is of great significance for monitoring physiological state and preventing and diagnosing diseases.

To date, many analytical techniques have been applied to detect neurotransmitters, including nanopore, chemiluminescence, spectrophotometry, capillary electrophoresis, and high-performance liquid chromatography (HPLC) [8,9]. However, some of the aforementioned techniques require complex pretreatment processes, and some rely on expensive equipment or highly specialized operators. In contrast, electrochemical (bio)sensors have advantages such as simplicity of operation, low cost, high sensitivity, good selectivity, fast analysis and easy miniaturization, and more importantly, they can be used for in vivo monitoring [5]. Therefore, electrochemical (bio)sensors are considered one of the most promising approaches for clinical determination of neurotransmitters.

Electrochemical (bio)sensors are analytical tools composed of a recognition component, a transducer, and a signal readout system [10,11,12]. The unique recognition component determines that the sensor only responds to specific analytes. For electrochemical biosensors, the recognition components are bioactive substances such as enzymes, antigens, antibodies, nucleic acids, etc. Electrochemical (bio)sensors can sense the physicochemical or biological changes in the analytes on the working electrode surface and convert them into measurable electrical signals (current, potential, resistance, etc.) using the transducer, after which the analyte is analyzed by the electrical signal displayed on the signal readout system [13]. Electrodes play a crucial role in electrochemical (bio)sensors. Bare electrodes generally face the problems of poor sensitivity and selectivity. Therefore, the modification of electrodes has been widely explored [14,15]. Research has shown that modifying nanomaterials with high electrical conductivity, large specific surface area and good electrocatalytic properties on the electrode surface can accelerate electron transfer on the electrode surface, amplify the output signal, and thereby improve the performance of electrochemical sensors [16,17,18]. For biosensors, modification of electrodes also promotes the immobilization of the biorecognition component [16,18].

Various nanomaterials, such as conducting polymers [3], metal nanostructures [19], carbon quantum dots [8], halloysite nanotubes [20], carbon nanofibers [21], carbon nanotubes [22], graphene and its derivatives [9], transition metal dichalcogenides [23,24], and some novel composites [25,26], have been successfully applied to fabricate electrochemical sensors for neurotransmitter detection. In recent years, MXene, a novel two-dimensional material, has received much attention. MXene has unique characteristics including large specific surface area, abundant surface functional groups, high electrical conductivity, good stretchability and biocompatibility [27,28,29], making it an ideal candidate for constructing electrochemical (bio)sensors. The applications of MXene-based electrochemical (bio)sensors in the detection of drugs, environmental pollutants, and disease biomarkers have been summarized [30,31,32,33,34,35]. The applications of MXene-based electrochemical (bio)sensors in neurotransmitter detection have been increasingly reported since 2018, but a systematic summary is lacking so far.

In this paper, the use of MXene-based electrochemical (bio)sensors for the detection of neurotransmitters including biogenic amines (DA, 5-HT, EP, NE), amino acids (Tyr), and soluble gases (NO and H_2_S) is systematically summarized (Figure 1), with a focus on the preparation strategies for MXene-based electrode materials. Meanwhile, the current problems of MXene-based electrochemical (bio)sensors for neurotransmitter detection are analyzed, and the prospects for this research field are presented. This paper aims to inform readers of the advances in MXene-based electrochemical (bio)sensors for neurotransmitter detection, and provide some helpful guidance for promoting the development of this field.

## 2. Brief Introduction to MXenes

### 2.1. Composition, Structure and Synthesis

Since the first MXene, Ti_3_C_2_T_x_, was discovered by Gogotsi and Barsoum et al. in 2011 [36], different kinds of MXene have been reported one after another. It is currently known that a MXene is a class of transition metal carbides, carbides or carbonitrides. MXene is mainly etched from the parent MAX phase. The MAX phase is a ternary layered compound with the general formula M_*n*+1_AX*_n_*, in which M stands for early transition metal elements (e.g., Ti, V, Cr, Zr, Nb, Mo), A is elements of groups IIIA~VIA (Al, Si, Ga, In etc.), X is carbon (C) and/or nitrogen (N), and *n* represents 1, 2, 3, or 4 [37,38,39]. Based on the fact that the M–X bond is stronger than the M–A bond and the high reactivity of etchants with the A element, the A atomic layer can be selectively etched off the MAX phase by some etchants to obtain MXene [40,41]. The resulting MXene can be expressed by the general formula M_*n*+1_X*_n_*T*_x_*, in which M and X represent the same elements as in the MAX phase, and T*_x_* represents the surface functional groups, such as –OH, –O, –F, and –Cl, that are introduced to the MXene surface during the etching process. With different values of *n*, the MAX phase can be mainly classified into four typical types: M_2_AX_1_ (211), M_3_AX_2_ (312), M_4_AX_3_ (413) and M_5_AX_4_ (514), and the etched MXene is correspondingly classified into four types according to its molecular structure: M_2_XT_x_, M_3_X_2_T_x_, M_4_X_3_T_x_ and M_5_C_4_T_x_ [37]. Figure 1 shows the constituent elements and typical molecular structures of the MAX phase and the etched MXene.

In terms of synthesis, MXene can be obtained from the MAX phase using different etching methods, such as HF etching, acid/fluoride salt etching (in-situ HF etching), alkali etching, molten salt etching, electrochemical etching, etc. [42]. The etching method determines the species of surface functional groups introduced to the MXene surface [43]. These methods of obtaining MXene by etching its MAX phase precursor can be summarized as “top-down” methods. Additionally, it is also feasible to prepare MXene by combining various atoms using some methods, which are referred to as “bottom-up” methods, such as chemical vapour deposition (CVD), template method, plasma-enhanced pulsed laser deposition (PLD), sputtering and pyrolysis [42,44]. The simple “top-down” methods are currently the dominant techniques for preparing MXene. However, both HF etching and fluoride-free etching inevitably introduce some functional groups and defects onto the surface of MXene. In contrast, the “bottom-up” methods are becoming increasingly attractive as they can address the above issues and enable more precise regulation of the designable MXene structures [44,45]. To date, more than 100 MXenes have been reported, of which more than 30 MXenes were synthesized experimentally and about 80 MXenes were predicted by theoretical calculations [46].

### 2.2. Properties, Delamination and Surface Modification/Functionalization Strategies

MXene has several advantages such as large specific surface area, rich surface functional groups, good biocompatibility, and excellent electronic, mechanical, and physicochemical properties, which make MXene an ideal candidate for manufacturing electrochemical (bio)sensors [27,28]. However, there are still two main problems with 2D MXene: on the one hand, the inherent aggregation and self-restacking of MXene layers not only affect their dispersity and stability, but also affect their electrochemical properties by reducing active sites and hindering electron transport [47]; on the other hand, the inherent surface functional groups of MXene cannot directly immobilize biorecognition elements through covalent bonds. Therefore, some strategies have been developed to address the above issues.

Delamination is a feasible strategy to alleviate the aggregation and self-restacking of MXene layers. Some cations and organic molecules can be used as intercalators to weaken the interaction forces between MXene layers, resulting in single- or few-layer MXene structures. The use of intercalation agents depends on the etching methods. With acid/fluoride salt etching, cations can be directly intercalated between adjacent MXene layers, forcing them to separate by sonication. When etched with HF, MXene layers can only be delaminated by adding certain cations or organic molecules (hydrazine [N_2_H_4_], urea, dimethyl sulfoxide [DMSO], isopropylamine [i-PrA], tetramethylammonium hydroxide [TBAOH], etc.) [33,48]. It is worth noting that the choice of intercalation agents is also related to the composition of the MXene; for example, DMSO is suitable for Ti_3_C_2_T_x_, but not for other MXenes, whereas i-PrA is suitable for Ti_3_C_2_T_x_, Nb_4_C_3_T_x_ and Nb_2_CT_x_ [35,48]. Surface modification/functionalization is another strategy to improve the performance of the MXene. Several hydrophilic polymers such as polyethylene glycol (PEG), polyvinylpyrrolidone (PVP) and soybean phospholipid (SP) have been utilized to modify the surface of MXenes, successfully improving the dispersion stability of MXenes under physiological conditions [48,49,50]. Some functional groups, such as phenylsulfonic and sulfonic groups (–SO_3_H) and carboxyl groups (–COOH), have been introduced to the surface of MXene to enhance its performance in specific applications (dye adsorption, heavy metal ion adsorption, etc.) [51]. For biosensors, amino groups (–NH_2_) have also been introduced onto the MXene surface via (3-Aminopropyl)triethoxysilane (APTES) to enhance sensing capability [49,51,52]. In addition, the introduction of other active materials into MXenes to prepare composites (or heterojunctions) is also a modification strategy. For example, metal nanoparticles, metal oxides, carbon nanotubes (CNTs), graphene, and conducting polymers have been introduced into MXenes to meet the requirements of various applications [30,53,54].

## 3. MXene-Based Electrochemical (Bio)Sensors for Neurotransmitter Detection

To date, a certain number of MXene-based electrochemical (bio)sensors have been constructed for the detection of neurotransmitters. The analytes detected by these (bio)sensors can be grouped into three categories: biogenic amines (DA, 5-HT, EP, NE), amino acids (Tyr), and soluble gases (NO, H_2_S). Various analytical methods have been applied to detect the analytes, including amperometry, differential pulse voltammetry (DPV), square wave voltammetry (SWV), and cyclic voltammetry (CV). Table 1 summarizes the MXene-based electrochemical (bio)sensors used for neurotransmitter detection, including analyte, working electrode, etchant for MXene, analytical method, performance and real sample for analysis. Some representative electrochemical neurotransmitter (bio)sensors are highlighted in the following text.

### 3.1. Biogenic Amines

#### 3.1.1. DA

DA is the most abundant biogenic amine neurotransmitter in the central nervous system. Most MXene-based electrochemical neurotransmitter sensors are fabricated for DA detection. For healthy individuals, the concentration of DA in the body usually ranges between 0.01–1 μM [9]. Abnormal DA levels are closely associated with many neurological and physiological diseases, such as Parkinson’s disease, schizophrenia, social anxiety, and addictive behavior [3,9]. DA generally coexists with uric acid (UA) and ascorbic acid (AA) in real biological fluids, and shares a similar oxidation potential window with them. In order to distinguish AA and UA in DA detection, various nanomaterials are needed to modify the electrode surface to improve the selectivity of electrochemical sensors.

A simple DA sensor was developed by modifying Nafion-stabilized Ti_3_C_2_T_x_ MXene on a glassy carbon electrode (GCE) [55]. The negatively charged surface functional groups of Ti_3_C_2_T_x_ promoted selective accumulation of positively charged DA. The Nafion-stabilized Ti_3_C_2_T_x_/GCE-based sensor showed much better electrochemical performance than that of a reduced graphene oxide (rGO)-based sensor under similar experimental conditions. A mixed phase MXene (Ti–C–T_x_) was synthsized and then deposited on GCE for DA analysis [56]. The Ti–C–T_x_/GCE showed excellent electrocatalytic activity and could clearly distinguish AA and UA in DA analysis. The Ti–C–T_x_/GCE-based sensor achieved a low limit of detection (LOD) for DA (0.06 µM), AA (4.6 µM), and UA (0.075 µM), and therefore can be used for the determination of DA in human urine samples. Nitrogen and sulfur co-doped Nb_2_C MXene (NS-Nb_2_C) was synthesized using thiourea as N,S resource [58]. The introduction of N,S heteroatoms into MXene enlarged the interlayer distance of MXene, increased its surface area, exposed more active sites and enhanced the electrical conductivity. The NS-Nb_2_C/Nafion/GCE-based sensor was satisfactory in detecting DA in simulated gastric juice. Ethylene diamine (EDA) aminated Nb_2_CT_x_ MXene was prepared to modify carbon cloth (CC) for DA detection [59]. The sensor based on EDA@Nb_2_CT_x_/CC obtained an ultra-low LOD of 300 pM. A sensor for simultaneous determination of DA and Tyr was prepared based on a Ti_3_C_2_ layer modified screen-printed electrode (Ti_3_C_2_/SPE) [60]. The DPV curve showed two clearly separated oxidation peaks corresponding to DA and Tyr, respectively. The Ti_3_C_2_/SPE sensor was able to simultaneously detect DA and Tyr in DA tablets and urine samples.

Introducing metals or metal oxides/sulfides/phosphides into MXene can yield MXene-based composite electrode materials with enhanced electrochemical properties. Ti_3_C_2_T_x_/Pt nanoparticles (PtNPs) were synthesized by reducing H_2_PtCl_6_ on the surface of accordion-like Ti_3_C_2_T_x_ [61]. The Ti_3_C_2_T_x_/PtNPs/GCE-based sensor achieved a good response for electrochemical analysis of DA and UA due to the intercalated PtNPs preventing aggregation of the MXene layer and facilitating electron transfer. Ti_3_C_2_T_x_/DNA/Pd/Pt nanocomposites were prepared for DA detection [62]. DNA was used to induce the in-situ growth of the Pd/Pt nanoparticles, as it can be adsorbed on the surface of Ti_3_C_2_T_x_ via π–π stacking. The introduction of Pd/Pt nanoparticles into Ti_3_C_2_T_x_ significantly improved the electrocatalytic activity. In another work, a flexible sensor based on a Au–Pd/Ti_3_C_2_T_x_/LSGE (laser-scribed graphene electrode) was fabricated [63]. A 3D porous LSGE was first prepared on a PI film, then Ti_3_C_2_T_x_ was cross-linked with LSGE via a C–O–Ti covalent bond to form the Ti_3_C_2_T_x_/LSGE, and finally Au–Pd nanoparticles were assembled on the Ti_3_C_2_T_x_/LSGE through a self-reduction process (Figure 2). As the large specific surface area and excellent electrochemical properties of Au–Pd/Ti_3_C_2_T_x_/LSGE enhanced the current response, the as-prepared sensor exhibited low LOD for DA (0.13 µM), AA (3 µM) and UA (1.47 µM). In addition, the sensor was successfully applied to detect DA, AA, and UA in urine samples, and to monitor their variation in sweat samples. Single-atom catalysts (SACs) with maximum atomic utilization rate have better catalytic activity compared to metal nanoparticles. Fe-SACs/Ti_3_C_2_T_x_ composites were synthesized to analyze DA and its end products VMA (vanillylmandelic acid) and HVA (homovanillic acid) [64]. The Fe-SACs/Ti_3_C_2_T_x_ composites exhibited excellent electrocatalytic activity for the oxidation of DA, VMA, and HVA. The sensor based on Fe-SACs/Ti_3_C_2_T_x_/LSGE was feasible for analysis of DA, VMA, and HVA in urine and serum samples, as well as for real-time monitoring of in situ cellular DA productions.

ZnO is widely used for the preparation of electrochemical sensors due to its good electronic properties and high chemical stability. ZnO/Ti_3_C_2_T_x_ composites were synthesized by the self-assemblling of amino functionalized ZnO and Ti_3_C_2_T_x_ MXene. On this basis, ZnO/Ti_3_C_2_T_x_/Nafion/Au electrodes were prepared for electrochemical detection of DA [65]. SnO_2_ quantum dots-functionalized Ti_3_C_2_ nanocomposites (SnO_2_QDs@Ti_3_C_2_ MXene) were synthesized using an in situ growth method for eletrochemical analysis of DA [66]. As the ultra-small SnO_2_QDs could be intercalated between adjacent Ti_3_C_2_ nanosheets, effectively preventing the restacking of Ti_3_C_2_, the electroactive surface area and electron transfer mobility of the obtained composite electrode materials were significantly increased. The SnO_2_QDs@Ti_3_C_2_-based sensor achieved an excellent electrochemical response to DA, with a LOD of 2.0 nM. Combing metal sulfides with MXene is a feasible method for preparing high-performance composite electrode materials. Nb_2_C/ZnS (mass ratio 1:1) nanocomposites were synthesized for the electrochemical analysis of DA [67]. A GCE modified with Nb_2_C/ZnS nanocomposites (Nb_2_C/ZnS/GCE) showed high selectivity for DA analysis. In another work, Nb_2_CT_x_@MoS_2_ heterostructures were synthesized using a hydrothermal method and their composition was optimized by setting different MoS_2_ contents. These Nb_2_CT_x_@MoS_2_ heterostructures were then modified on CC to detect DA [68]. Due to the interaction between MoS_2_ and Nb_2_CT_x_, the Nb_2_CT_x_@MoS_2_/CC-based sensor achieved ultrasensitive electrochemical detection of DA, with a wide linear range (1 fM–100 nM) and an ultra-low LOD of 0.23 fM. Bimetallic phosphides with metalloid characteristics and good electrocatalytic activity are more suitable for constructing electrochemical sensors. A highly sensitive electrochemical sensor for DA detection was fabricated based on a CC/Ti_3_C_2_T_x_/NiCoP composite [69]. The CC/Ti_3_C_2_T_x_/NiCoP composite was synthesized using a hydrothermal method and subsequent phosphating process. Combing the large specific surface area of NiCoP quantum dots, the abundant active sites of porous NiCoP nanowires, the good hydrophilicity of Ti_3_C_2_T_x_ nanosheets and the excellent conductivity of CC substrates, the sensor achieved a wide linear range from 0.17 μM to 784.55 μM and a low LOD of 0.18 nM.

Combing carbon nanomaterials such as carbon nanotubes (CNTs), graphene (Gr) and its derivatives with MXene is also a commonly used method for preparing high-performance MXene-based composite electrode materials. Bilayer Ti_3_C_2_T_x_/rGO heterostructures were synthesized through electrochemical reduction for electrochemical detection of DA and UA [70]. Ti_3_C_2_T_x_ and electrochemically reduced holey graphene modified glass carbon electrodes (Ti_3_C_2_T_x_-ERHG/GCE) were prepared for DA detection. With the highly nanoporous ERHG hindering the agglomeration and oxidation of Ti_3_C_2_T_x_, Ti_3_C_2_T_x_ reducing the non-specific adsorption, and their synergistic effect promoting the electron transfer rate, the Ti_3_C_2_T_x_-ERHG/GCE-based sensor exhibited excellent electrochemical performance, anti-biofouling properties, and good sensitivity in the detection of DA in serum and artificial cerebrospinal fluid (aCSF) [71]. Later, single-layer MXene and holey graphene (S-Ti_3_C_2_T_x_/HG) nanocomposites were further prepared by the same research group for electrochemical sensing of DA [72]. A highly sensitive DA electrochemical sensor was fabricated based on Ti_3_C_2_ MXene, graphitized multi-walled carbon nanotubes and a ZnO nanosphere-modified glassy carbon electrode (Ti_3_C_2_/G-MWCNTs/ZnO/GCE) [73].

Conductive polymers and organic compounds can be used as effective regulators to improve the electrochemical ability and structural stability of 2D MXene. Three polymers with good biocompatibility and hydrophilicity, PVP (polyvinylpyrrolidone), PVA (polyvinyl alcohol) and PAM (polyacrylamide), were employed to modify Ti_3_C_2_ MXene in order to prepare stable layered Ti_3_C_2_-PVP, Ti_3_C_2_-PVA and Ti_3_C_2_-PAM nanocomposites for electrochemical detection of DA [74]. Ti_3_C_2_T_x_/PPy (polypyrrole) nanocomposites were synthesized for detection of DA and UA [75]. Poly(3,4-ethylenedioxythiophene):poly(styrenesulfonate) (PEDOT:PSS) was integrated with MXene via electropolymerization to construct a DA sensor [76]. PDI (perylene diimide) was self-assembled with MXene via H-bonding to synthesize PDI-Ti_3_C_2_T_x_ composites [77]. Graphitic pencil electrodes (GPE) with sp^2^-hybridized carbon were then modified with PDI-Ti_3_C_2_T_x_ composites for electrochemical analysis of DA. Due to the synergistic effect of MXene and PDI resulting in a highly sensitive sensing interface, and the electrostatic attraction between negatively charged PDI and positively charged DA facilitating the adsorption of DA, the PDI-Ti_3_C_2_T_x_/GPE-based sensor achieved superior sensitivity and selectivity, and also exhibited a good response in analyzing DA in human serum samples. Metal organic frameworks (MOFs) have been promising materials in sensing applications because of their tunable porosity and excellent adsorption performance. 2D/2D NiCo-MOF/Ti_3_C_2_ heterojunctions were prepared and coated on a carbon cloth electrode (CCE) for the electrochemical determination of DA, UA and AP (acetaminophen) [78]. Integrating the abundant catalytic active centers of NiCo-MOF and the superior electrical conductivity of Ti_3_C_2_ MXene, the NiCo-MOF/Ti_3_C_2_/CCE-based sensor exhibited good performance for the oxidation of DA, UA and AP, with completely separated signal peaks. A Ti_3_C_2_ film was synthesized by hydrogen bonding self-assembly of UIO-66-NH_2_ and Ti_3_C_2_ [79], and its electrochemical performance was optimized by adjusting the mass ratio of Ti_3_C_2_ and UIO-66-NH_2_. Benefiting from its large specific area, excellent electronic conductivity and water dispersibility, the Ti_3_C_2_ film exhibited strong selectivity and sensitivity to DA.

In a study [80], Ti_3_C_2_Cl_2_ MXene was prepared using molten salt etching. Ionic liquid (IL) was then applied to stabilize the inherently unstable Ti_3_C_2_Cl_2_ and enhance its electrocatalytic capability. As the strong π–π interaction between IL and Ti_3_C_2_Cl_2_ inhibited leaching of the electrode material and promoted electron transfer at the electrode interface, the IL-Ti_3_C_2_Cl_2_/GPE-based sensor showed robust activity towards DA oxidation. Notably, this is the first time that MXene was prepared using molten salt etching for neurotransmitter detection.

In addition to active nanomaterials, enzymes can also be used to catalyze the oxidation of DA. A laccase-Ti_3_C_2_-LIG electrode was prepared by immobilizing laccase on a Ti_3_C_2_ modified laser-induced graphene electrode. The laccase-Ti_3_C_2_-LIG electrode was then encapsulated in a 3D printed microfluidic device for electroanalysis of DA. With the ideal matrix provided by the Ti_3_C_2_ modified LIG network for laccase immobilization, as well as the stable environment provided by the microfluidic device to maintain laccase sensitivity, the resulting laccase-Ti_3_C_2_-LIG-based sensor exhibited good analytical performance in real-time detection of blood serum and synthetic urine samples, and demonstrated good potential in point-of-care applications [81].

Changing the 2D structure of MXene to other dimensional nanostructures is also an effective way to prevent the restacking or aggregation of MXene, thereby improving its electrochemical performance. In one study, the positively charged surfactant DODA (dimethyldioctadecylammonium bromide) was inserted between negatively charged MXene nanosheets through electrostatic interaction [82]. DODA not only prevented the restacking or aggregation of MXene nanosheets, but also combined with MXene to form an organic soluble MXene/DODA complex, which was further utilized to fabricate a 3D porous MXene nanostructure via the breath figure method (Figure 3). As the porous nanostructure of MXene provided more active sites and facilitated mass transport, the 3D porous MXene-modified indium tin oxide (ITO) electrode achieved good electrochemical performance for DA detection, with a low LOD of 36.8 nM, and showed good accuracy for DA determination in human urine samples. In another study, a 3D-printed nanocarbon electrode (3DE) was fabricated with graphene/polylactic acid (PLA) filament using the fused deposition modeling (FDM) technique. The 3DE was then modified with synthesized MXene quantum dots (MQDs) to construct an electrochemical sensor for the determination of DA [83].

#### 3.1.2. 5-HT

5-HT is an inhibitory neurotransmitter in the central nervous system that plays an important role in regulating calm physiological behaviors such as learning, sleeping, and emotions [95]. 5-HT deficiency can lead to mental disorders, such as intellectual retardation, sleep disorders, depression, and anxiety [3,95].

In one study, l-cysteine-terminated triangular silver nanoplates (Tri-AgNP/l-Cys) were loaded onto Ti_3_C_2_T_x_ MXene to prepare Tri-AgNP/l-Cys/Ti_3_C_2_T_x_ for electrochemical detection of 5-HT [84]. The preparation process for the Tri-AgNP/l-Cys/Ti_3_C_2_T_x_ electrode material is shown in Figure 4. Since l-Cys with sulfhydryl group forms a more stable Ag–S bond with silver, it can selectively replace trisodium citrate (TSC) in a TSC-terminated triangular silver nanoplate (Tri-AgNP/TSC) to obtain Tri-AgNP/l-Cys. The Ti_3_C_2_T_x_ MXene, with large surface area and good biocompatibility, provided a good loading platform for Tri-AgNP/l-Cys, while the intercalation of Tri-AgNP/l-Cys avoided the restacking of Ti_3_C_2_T_x_ nanosheets. Under optimized conditions, the Tri-AgNP/l-Cys/Ti_3_C_2_T_x_/GCE-based sensor exhibited good performance for 5-HT detection, with a LOD of 0.08 μM. An electrochemical sensor based on a Ti_3_C_2_T_x_ MXene/single-walled carbon nanotubes (SWCNTs) nanocomposite was developed for dynamic monitoring of 5-HT secretion from cultured living cells [85]. Ti_3_C_2_T_x_-rGO nanocomposites were synthesized by hydrazine reduction and subsequent ultrasonic treatment [86]. The combination of rGO and Ti_3_C_2_T_x_ MXene alleviated the aggregation of MXene nanosheets and exposed more effective active sites, resulting in better electrochemical performance. The Ti_3_C_2_T_x_-rGO/GCE-based sensor exhibited high sensitivity for 5-HT detection, and could successfully be used to detect 5-HT in blood plasma.

#### 3.1.3. EP

EP, also known as adrenaline, is an excitatory neurotransmitter synthesized in the adrenal gland and released into the blood in emergency and stressful situations [2,96]. EP regulates the heartbeat, and also plays an important role in the treatment of cardiac arrest, hypertension, asthma, severe allergic reactions, and other allergic diseases [3,96].

A Ti_2_CT_x_ MXene/graphite composite paste electrode (MXene/GCPE) was fabricated for the detection of EP [87]. This electrode can clearly distinguish EP, AA, and 5-HT in phosphate buffer solution (PBS), and can be used to detect EP in pharmaceutical samples. 3D reduced graphene oxide/Ti_3_C_2_T_x_ MXene (rGO/Ti_3_C_2_T_x_) was synthesized by freeze-drying and heat treatment and transferred to indium tin oxide (ITO) for electrochemical detection of EP [88]. In another work, Ti_3_C_2_T_x_ MXene/N-doped reduced graphene oxide (MXene/N-rGO) composites were synthesized via a simple hydrothermal reaction using ethylenediamine (EDA) as a GO reducing agent and nitrogen source [89]. The composites achieved enhanced electrocatalytic activity towards EP due to the synergistic effect of Ti_3_C_2_T_x_ MXene and N-rGO.

Molecularly imprinted polymers (MIP) not only have excellent selective recognition and adsorption capabilities for target molecules, but also have higher stability compared to other biological recognition components. As a result, molecularly imprinted sensors have attracted intense interest in recent years. A molecularly imprinted sensor for electrochemical determination of EP was developed using MXene/carbon nanohorns (CNHs) as electrode modification material, EDOT-CH_2_OH as functional monomer, and EP as template molecule [90]. The preparation process for the pEDOT-CH_2_OH-MIP/MXene/CNHs/GCE is shown in Figure 5. Due to the high conductivity, large specific surface area, and excellent electrocatalytic ability of MXene/CNHs, as well as the specific recognition ability of stable pEDOT-CH_2_OH-MIP for target EP molecules, the sensor based on pEDOT-CH_2_OH-MIP/MXene/CNHs/GCE displayed excellent sensing performance, with a wide linear range (1.0 nM to 60.0 μM) and a low LOD (0.3 nM). The sensor also demonstrated satisfactory feasibility in urinary EP analysis.

#### 3.1.4. NE

NE is a biogenic amine neurotransmitter secreted by the adrenal medulla [3]. Abnormal NE concentration is associated with various diseases, such as neuroblastoma, Parkinson’s diseases, multiple sclerosis, coronary heart disease, depression, etc. [3,96].

Titanium dioxide/MXene with polyvinyl alcohol/graphene oxide (TiO_2_/MXene-PVA/GO) hydrogel was successfully prepared and modified on a screen-printed carbon electrode (SPCE) by repeated freezing–thawing treatment for urinary NE determination (Figure 6). As the combination of TiO_2_ and MXene enhanced the conductivity, electrocatalytic activity and stability of the electrode, and the PVA/GO hydrogel increased the specific surface area and sample absorption, the sensor based on TiO_2_/MXene-PVA/GO/SPCE showed good electrochemical response and sensitivity, and could be used for the determination of NE in urine samples. In addition, the sensor can be integrated into a pantyliner for offline urinary NE determination, suggesting strong potential for preparing wearable sensors for point-of-care testing [91].

### 3.2. Amino Acids

Tyr, one of the major amino acid neurotransmitters in the human nervous system, is used as a precursor to three important biogenic amine neurotransmitters (DA, EP and NE). Abnormal levels of Tyr are closely associated with various diseases, such as dementia, Parkinson’s disease, depression, anxiety, etc. [3,97]. As shown in Figure 7, an electrochemical sensor for Tyr detection was developed based on Ti_3_C_2_T_x_/CNTs/CuMOF composite-modified GCE [92]. The intercalation of CNTs effectively prevented the aggregation of MXene nanosheets, and the introduction of porous CuMOF octahedral particles facilitated the adsorption of Tyr, thereby improving the electrocatalytic property of the electrode. Based on the rational design strategy of the electrode materials, the as-prepared sensor achieved a wide detection range (0.53 μM–232.46 μM) and a low LOD (0.19 μM), and can be used for Tyr detection in human serum.

### 3.3. Soluble Gases

Some soluble gases have been identified as gasotransmitters. NO is a gasotransmitter involved in various neurological functions including sleep and appetite regulation, neurotransmission, learning and memory [3]. In one study, MOF@V_2_CT_x_ heterostructures were prepared by partial in-situ conversion of V_2_CT_x_ MXene to MOF for electrochemical NO sensing [93]. Due to the metalloid nature of V_2_CT_x_ MXene, charge redistribution at the heterojunction interface facilitating charge transfer, and enhanced NO adsorption energy at the heterojunction interface, the MOF@V_2_CT_x_ heterostructures exhibited significantly improved electrochemical NO sensing performance compared to those of bare MXene and MOF.

H_2_S is another endogenous gasotransmitter that is widely distributed in the body and plays an important role in neurotransmission, antioxidation, and vascular tone regulation [94,98]. Abnormal H_2_S concentrations may be related to neurodegeneractive disorders, gastrointestinal diseases, diabetes and cancer [98,99]. A direct electrochemical sensor based on Ti_3_C_2_T_x_ MXene-modified GCE was fabricated for H_2_S detection [94], which exhibited good sensing performance with a low LOD of 16.0 nM and a wide detection range from 100 nM to 300 µM. This work pioneered the use of MXene for H_2_S electrochemical sensing.

## 4. Conclusions and Outlook

This paper reviews the advances in MXene-based electrochemical (bio)sensors for neurotransmitter detection. First, the composition, structure, synthesis, properties, delamination and surface modification/functionalization of MXene are briefly described. Then, a systematic summary of MXene-based electrochemical (bio)sensors for the detection of neurotransmitters including biogenic amines (DA, 5-HT, EP, NE), amino acids (Tyr), and soluble gases (NO and H_2_S) is presented, focusing on the preparation strategies for electrode materials. In the past few years, research in this field has been dedicated to improving the sensitivity, selectivity, and stability of electrodes. Thus, MXene-based electrode materials with high electrical conductivity, large specific surface area, porous structure, good biocompatibility, and anti-fouling properties have been prepared by combining metal nanoparticles, metal oxides/sulfides/phosphides, carbon nanomaterials, conductive polymers, MOFs, etc., with MXene or by changing the 2D structure of MXene to other dimensional nanostructures. In view of this, significant progress has been made in MXene-based neurotransmitter electrochemical sensors. However, MXene-based neurotransmitter electrochemical sensors still face some deficiencies and need further development and improvement: (1) MXene is mainly prepared using HF etching and acid/fluoride salt etching. The residual –F on the surface of MXene affects its electrochemical performance. It is necessary to develop simple fluorine-free preparation methods to achieve large-scale production of MXene in the future. (2) The species of neurotransmitters detected by MXene-based electrochemical sensors are limited; some important species, such as histamine, acetylcholine, aspartic acid, and glutamate, have not been investigated. (3) A certain human disease or behavior is generally the result of coordinated effects of multiple neurotransmitters; therefore, it is necessary to simultaneously detect the relevant neurotransmitters. At present, simultaneous detection is still in its infancy, and thus it is of great significance to further develop electrochemical sensors for simultaneous detection of several neurotransmitters. (4) MXene-based neurotransmitter electrochemical sensors have been developed for detecting neurotransmitters in biological fluids (e.g., serum and urine), some of which have shown potential for POC detection. Next, it is meaningful to further implement their POC applications by developing the relevant devices. (5) MXene-based neurotransmitter electrochemical sensors have not yet achieved real-time detection in the human body; therefore, the future direction is to develop MXene-based neurotransmitter electrochemical sensors that can be used for real-time detection in the human body. It is also necessary to develop integrated, miniature, and implantable sensors for neurotransmitter detection in blood, as well as wearable flexible sensors for urinary neurotransmitter detection. These sensors can be combined with miniature electrochemical workstations and wireless sensing devices or mobile phones to ultimately enable real-time and long-term monitoring of human neurotransmitters. In summary, the research on MXene-based neurotransmitter electrochemical sensors is still at a preliminary stage, and there is still a long way to go to truly achieve their clinical applications. This paper presents the design strategies, current bottlenecks and future directions for MXene-based neurotransmitter electrochemical sensors, aiming to provide some reference for further development in this field.

## Data Availability

Not applicable.

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
