# Peer review of "Advances in MXene-Based Electrochemical (Bio)Sensors for Neurotransmitter Detection"

_micromachines, 2023, doi:10.3390/mi14051088_

Round 1

Reviewer 1 Report

The article “Advances of MXene-based Electrochemical Biosensors for Neurotransmitter Detection” is relevant because the MXenes materials have been growing up so fast in the scientific community and these materials are very interesting for electrochemical applications, for this, I recommend the review for publication in Micromachines. I have comments, questions and suggestions that can enhance the performance of manuscript, enumerated below:

1.     I suggest authors change the word ‘biosensor’ for (bio)sensor, because every biosensor is a sensor too. But every sensor is not a biosensor. Then, the nomenclature (bio)sensor is more appropriated.

2.     Keywords must be different from the title to get more visibility.

3.     Authors give a definition of biosensor, but review is not about just biosensors. I think this paragraph should be changed with a different perspective.

4.     Authors should include some references in detection of neurotransmitters in fourth paragraph, such as: 10.3390/bios13030308; 10.1016/j.bej.2022.108565; 10.1016/j.microc.2021.107141; 10.1016/j.diamond.2023.109677; 10.1016/j.compositesb.2019.107649;

5.     In the last paragraph of Introduction section, authors emphasized that “This paper systematically summarizes the application of MXene-based electrochemical biosensors for neurotransmitter detection, with a focus on the strategies to improve the performance of MXene-based electrode materials.” But there is a lot of examples where there is sensor and not a biosensor . for this, I think authors should change for (bio)sensors, and then it gets more general. Authors should verify the definition of a biosensor in IUPAC recommendations, such as: https://doi.org/10.1515/pac-2018-0109

6.     Conclusion is confusing. Authors should improve. (without mention of Figures)…

Author Response

Reviewer #1:
(1) I suggest authors change the word ‘biosensor’ for (bio)sensor, because every biosensor is a sensor too. But every sensor is not a biosensor. Then, the nomenclature (bio)sensor is more appropriated.

Response: Thanks for your suggestion. We have changed the word ‘biosensor’ to (bio)sensor in the revised manuscript.

(2) Keywords must be different from the title to get more visibility. 

Response: The keywords have been modified to get more visibility in the revised manuscript.
(3) Authors give a definition of biosensor, but review is not about just biosensors. I think this paragraph should be changed with a different perspective.

Response: Thank you for your careful review. The third paragraph of the Introduction has been modified according to your suggestion.

(4) Authors should include some references in detection of neurotransmitters in fourth paragraph, such as: 10.3390/bios13030308; 10.1016/j.bej.2022.108565; 10.1016/j.microc.2021.107141;

10.1016/j.diamond.2023.109677; 10.1016/j.compositesb.2019.107649.

Response: The above relevant references have been added to the fourth paragraph of the manuscript.

(5) In the last paragraph of Introduction section, authors emphasized that “This paper systematically summarizes the application of MXene-based electrochemical biosensors for neurotransmitter detection, with a focus on the strategies to improve the performance of MXene-based electrode materials.” But there is a lot of examples where there is sensor and not a biosensor . for this, I think authors should change for (bio)sensors, and then it gets more general. Authors should verify the definition of a biosensor in IUPAC recommendations, such as: https://doi.org/10.1515/pac-2018-0109.

Response: Thanks for your suggestion. We have redefined the electrochemical (bio)sensors in the third paragraph of the manuscript.

(6) Conclusion is confusing. Authors should improve. (without mention of Figures)…

Response: We have made modifications to the conclusion by removing its summary figure, and adding a scheme to describe the content of this article in the Introduction section.

Reviewer 2 Report

This paper reviews the advances of MXene-based electrochemical biosensors for neurotransmitter detection. This review is informative so that it is worth of publication in Micromachines. However, the following minor concerns should be addressed before publication.

1. It is better to have a scheme to describe the contents in introduction. For example, in the scheme, there should be classification of neurotransmitters (top); a brief illustration of electrochemical biosensor is also needed (middle); a zoomed-in figure showing the material used in electrodes. At the same time, Figure 8 which summarizes the whole paper can be used as Table of content (TOC).  

2. In the third paragraph of introduction, the authors describe “Electrodes play a crucial role…”, then the following sentence says “the modification of nanomaterials…”. Why nanomaterials? What about normal substrates? There is no rationale, or one sentence is needed as transition (maybe some references as well).

3. Some of the figures are in low resolution, such as Figure 5 and 6. Please consider using better quality figure by downloading high-resolution figures.

4. Please refine the abbreviations in the manuscript and make sure to use full name when mentioning the concept for the first time. For example, dopamine (DA) needs to be abbreviated in the first paragraph of introduction. The following paragraphs as well as table can use DA instead of the full name. Same for epinephrine, norepinephrine, and so on.

5. Please use the correct format for all the units. For example, in the table and main text, some of the “uM” units should be “µM”. Please check the whole manuscript.

The English is easy to read. Minor English language editing is required.

Author Response

Reviewer #2:
(1) It is better to have a scheme to describe the contents in introduction. For example, in the scheme, there should be classification of neurotransmitters (top); a brief illustration of electrochemical biosensor is also needed (middle); a zoomed-in figure showing the material used in electrodes. At the same time, Figure 8 which summarizes the whole paper can be used as Table of content (TOC).

Response: Thanks for your suggestion. We have added a scheme in the Introduction part to illustrate the content of this article and removed Figure 8 from the conclusion section in the revised manuscript.
(2) In the third paragraph of introduction, the authors describe “Electrodes play a crucial role…”, then the following sentence says “the modification of nanomaterials…”. Why nanomaterials? What about normal substrates? There is no rationale, or one sentence is needed as transition (maybe some references as well).

Response: Because the normal bare electrodes generally face the problems of poor sensitivity and selectivity. Therefore, the modification of nanomaterials on electrodes is an effective way to improve the performance of electrochemical sensors. We have made modifications to the third paragraph in the revised manuscript.
(3) Some of the figures are in low resolution, such as Figure 5 and 6. Please consider using better quality figure by downloading high-resolution figures.

Response: As the original Figures 5 and 6 are not very clear, we have made modifications to them. The modified Figures 5 and 6 are shown below.

Figure 5. Preparation steps and EP determination of pEDOT-CH2OH-MIP/MXene/CNHs/GCE. Reprinted with permission from [90].

Figure 6. Preparation process of TiO2/MXene-PVA/GO/SPCE and its off-line urinary NE detection. Reprinted with permission from [91].

(4) Please refine the abbreviations in the manuscript and make sure to use full name when mentioning the concept for the first time. For example, dopamine (DA) needs to be abbreviated in the first paragraph of introduction. The following paragraphs as well as table can use DA instead of the full name. Same for epinephrine, norepinephrine, and so on.

Response: Thank you for your suggestion. The abbreviation has been modified in the revised manuscript based on your suggestion.

(5) Please use the correct format for all the units. For example, in the table and main text, some of the “uM” units should be “µM”. Please check the whole manuscript.

Response: Thank you for carefully review. We have carefully checked the format of all units in the whole manuscript and made some modifications.